# Nursing Assistant in Italy: The Principle of Delegation of Health Activities and Liability Profiles

**DOI:** 10.3390/nursrep15120443

**Published:** 2025-12-11

**Authors:** Livio Pietro TRonconi, Vittorio Bolcato, Luca Bianco Prevot, Giuseppe Basile

**Affiliations:** 1Department of Health and Life Science, European University of Rome, 00163 Rome, Italy; 2Maria Cecilia Hospital, GVM Care & Research, 48033 Cotignola, Italy; 3Astolfi Associati Law Firm, 20121 Milan, Italy; 4Maria Beatrice Hospital, GVM Care & Research, 50121 Florence, Italy; 5Residency Program in Orthopaedics and Traumatology, University of Milan, 20122 Milan, Italy; 6Trauma Unit and Emergency Department, IRCCS Istituto Ortopedico Galeazzi, 20157 Milan, Italy; 7Department of Biomedical Sciences and Public Health, University “Politecnica delle Marche” of Ancona, 60124 Ancona, Italy

**Keywords:** nursing assistant, nursing, continuity of care, delegation, professional teams, health tasks, medico-legal

## Abstract

**Background/Objectives:** The institution of the nursing assistant (NA) profile in Italy, established by the Decree 28 February 2025 responds to the chronic shortage of healthcare personnel, especially in nursing. This figure, non-healthcare but trained to perform basic healthcare tasks, aims to support nurses and ensure continuity of care, especially in community and long-term care settings, through further nursing activities delegation. The model aligns with international practices, emphasizing delegation within multiprofessional teams, based on formalized protocols and continuous on-site training, within standardized, low-discretion contexts. The delegation of health activities, however, presents legal and medico-legal challenges regarding scope of practice and supervision. **Methods:** The aim of this paper is critically discussing delegation of health activities to non-healthcare providers and the related issues of liability in team-based delivery of care, considering the specific regulatory setting of health providers in Italy. **Results:** Critical activities such as nutrition and hydration, in particular artificially, and drug administration highlight the limits of delegation and the ongoing need for professional nurse oversight. In pre-hospital emergency care, standardized, protocol-based systems and simulation-based training successfully integrate non-healthcare personnel within the health team. Conversely, chronic and long-term care remain fragmented, lacking organizational maturity, regular supervision, and uniform regulation. **Conclusions:** The decree represents a step toward structured team-based care, but its success depends on robust governance, protocol clarity, organizational guidelines, and sustained practice-based education to prevent unsafe delegation and unauthorized practice.

## 1. Introduction

The establishment of the new role of nursing assistant (*Assistente Infermiere*) in Italy, introduced with the Decree of the President of the Council of Ministers (D.P.C.M.) of 28 February 2025, represents one of the responses to the growing difficulty, escalated after COVID-19 pandemic, faced by the national healthcare service in ensuring adequate staffing to provide comprehensive and continuous care across various hospital, primary care, and social-healthcare settings [1,2]. Certain care areas, more than others, are under greater pressure in terms of the deficit of healthcare professional’s workforce and expertise, such as primary care, home care and palliative care, and nursing homes and the territorial emergency/urgent care system, despite the launch of reforms concerning professional competences, including those of nurses [3,4,5,6]. It should also be noted that the right to health protection in Italy must be interpreted in a positive sense, as a duty of activation by the State and Regions through a service for the provision of care. This entails guaranteeing the quality of care and its continuity, governing both finance and organization, as well as of human capital, and thus setting up the required legislative intervention.

The institution in Italy of the nursing assistant is part of a socio-cultural and regulatory evolution, in line with international models, that has progressively redefined, starting from the more structured and standardized framework of hospital care, the roles and responsibilities of healthcare workers involved in care delivery, particularly those within the nursing profession [7,8]. Its purpose is to equip non-healthcare personnel with fundamental competencies and skills required for basic healthcare support. Such activities complement person-centered and first-aid tasks which, by their nature, lie outside the strictly clinical domain and pertain more broadly to general citizenship responsibility. In supporting care activities, both within emergency care and even more so in chronic diseases and elderly care management, there is a growing trend in leveraging nurse work burden, increasing professionalism, and thus ensuring continuity and comprehensive delivery of care [9,10]. This multiprofessional and team-based model in the ongoing Italian format, including non-healthcare personnel, is founded on the integration of targeted skills and training, enabling the delegation of selected healthcare activities in accordance with formalized and predefined care and organizational protocols [11].

Delegation is an extraordinarily complex construct, both to grant effective care delivery and legal accountability. Shore et al. highlighted three levels of factors involved. At the macro level, this is ensured by the consistency and clarity of government and state regulations; at the meso level, by the structured organization of processes and the definition of operative protocols and guidelines; and at the micro level, by the clear relationship between the delegator and the delegate within a team-based practice [12]. Crevacore et al. highlighted the five principles for effective delegation: assigning the right element, under the right circumstances, to the right person, with the right supervision, and through clear direction and communication [13]. All this while taking into account the space of autonomy in relation to delegated action, namely the possibility of adapting its content based on the context, strictly limited to licensed healthcare professionals. The delegation of health activities, as currently adopted in Italy, is usually achieved through entrusting providers with dedicated and generally non-university training pathways, within standardized care processes and clinically stabilized scenarios. Otherwise in clinical situations requiring the systematic application of procedures and flowcharts, such as acute trauma management, as no autonomy of judgment is possible for non-healthcare personnel [14]. This is therefore profoundly different from allied health personnel (AHP), who support medical staff both in clinical and administrative settings, as they are recognized as healthcare professionals by meeting the minimum requirements of education, regulation, and licensing. Allied health assistants (AHAs) on the contrary are positioned as non-healthcare equivalents to NA, increasingly taking on delegated tasks from allied health professionals. This redistribution of responsibilities allows highly qualified health practitioners to focus on patients with more complex needs. AHAs typically perform duties related to hygiene, educational support, posture management, and daily living activities, and their role is characterized by its task-oriented nature [15]. Regarding other supportive roles, such as caregivers or home aides, the situation is profoundly different in the Italian context. In fact, even if they address general living and assistance needs similar to those of a nursing home or residential facility, these figures operate at the user’s residence, typically outside the scope of care and jurisdiction of the health service and therefore not subject to the requirements, controls, and liability criteria applied to healthcare facilities or professions. They are generally non-healthcare, partly based on voluntary work, and usually lack even minimal healthcare training.

The aim of this manuscript is to critically examine the delegation of healthcare activities to non-healthcare providers and the related issues of liability within team-based models of care. This requires enhancing the value of training, the organizational context, and the type of care delivered, with the goal of clearly defining those areas of intervention entrusted to nursing assistants. Such activities should allow for an adequate—though secondary—level of supervision or control by the nurse, or be performed directly through standardized protocols, thereby appropriately empowering nursing assistants and enabling nurses to focus on more advanced professional functions.

## 2. The Nursing Assistant Role in Italian Healthcare Context

The new role, comparable to the certified nurse assistant (CNA) or healthcare operator in the Anglo-Saxon model, is defined as a non-healthcare member of the healthcare team who works under the supervision of licensed nurses [13]. They are operators trained in a non-academic setting, through training services under regional authority. They provide basic care and help patients with daily living activities within the processes of care delivery activated by the national health service.

In the Decree Annex, which entails the agreement for the establishment of the professional profile of the nursing assistant, the Government details the role and context of their practice: activities directed toward the patient and to meeting their daily nutritional, hygiene, and more general care needs, which are typical of an assistance-oriented role. English case law, though inconsistently, encompasses these within the notion of “basic care” [16]. The Italian text uses a comparable expression: healthcare-related assistance needs (Article 4, paragraph 3, *bisogni assistenziali di tipo sanitario*).

Then, depending on the clinical severity of the patient and the organizational context (primary care and hospital settings; social services; residential or semi-residential), the nursing assistant performs delegated activities which include those properly of a healthcare nature, such as support in artificial nutrition/hydration and devices management, administration of therapies for chronic treatments via natural routes, pain management and vital sign monitoring, as well as the performance of diagnostic tests such as electrocardiograms.

NAs are responsible for the accuracy and appropriateness of the activities performed, in particular those constitutive of the figure and not based on delegation (Article 1, paragraph 4) [17]. Moreover, they can perform managerial, organizational, and educational support (Article 1, paragraph 3).

The nurse, by contrast, is a healthcare professional who, following the 1999 reform and the comprehensive revision of professional bodies initiated in 2018, bases their autonomous care activities on the professional profile defined by the educational framework reached through university training, regulation, and licensing, and is therefore accountable accordingly [18]. Therefore, the nurse and the nursing assistant (NA) are referred to as a functional dyad. However, since one is a healthcare professional and the other is not, the care leadership primarily lies with the healthcare professional, resulting in a constitutive imbalance of responsibility between the two members of the dyad. The dyad’s effectiveness necessarily relies on the demarcation of responsibilities by “delegated activities”, within a context that is organizationally, training, or culture driven [19].

What is relevant and worth discussing are the ways in which the Italian national decree, following a model also present abroad, addresses the issue of the lawfulness of the delegation and performance of healthcare activities to the NA, in relation to potential unauthorized healthcare professional practice.

According to the Italian Decree, delegated healthcare tasks are to be performed within planned procedures and well-defined healthcare contexts, characterized by low-discretion situations and aimed at supporting the overall delivery of care (Article 4, paragraph 3) [20]. The methods of delegation include either direct indication by the nurse according to the planned procedures or the overall care plan, or according to formal case-by-case ad hoc assignment by the nurse (Article 4, paragraph 2) [21].

Moreover, this new model must take place within innovatively organized healthcare systems (preamble of the Annex 1), calling policymakers and healthcare managers to address care planning, workforce allocation, and the distribution of responsibilities [22].

## 3. Delegated Tasks, Supervision, and Medico-Legal Issues

Within the Italian context, certain critical issues emerge from a medico-legal standpoint concerning the lawfulness of the NA’s actions and the allocation of responsibilities in the cycle of performance of those healthcare and delegated activities. These providers are in fact, by definition, not endowed with discretionary capacity nor authorized to act autonomously within the healthcare field. However, some “assistance elements” are of particular significance as they are not limited to the execution of an activity but in some ways require discretionary effort in acting.

This becomes particularly evident when nursing supervision is involved, as a mechanism for verifying and guiding discretionary behavioral adjustments, especially in the pharmacotherapy path. Regarding supervision, the concept appears nuanced and open to divergent interpretations, similarly to what has already occurred in the past between physician and nurse, and nurse and another generic healthcare assistant [23]. Indeed, Italian case law on the issue has typically developed within the hospital setting and concerns the distinction between the provider’s necessary physical and simultaneous presence, versus a more general presence within the ward and availability in case of need or upon alert. However, in the less structured context of community-based services and nursing homes, the fluctuation in previous interpretations tends to favor a viable notion of mere on-call availability and comprehensive governance regarding the verification of the care plan implementation, otherwise an alternative and more restrictive approach would prove ineffective in responding to the limited presence of healthcare personnel, often linked to staff requirements plus structural staff shortages.

As for the management and use of artificial means of nutrition and hydration, such as intravenous drips or nasogastric tubes, this is particularly relevant in those facilities till the palliative care settings. These are delicate interventions that may nonetheless require even limited but additional discretionary actions of an assisting yet healthcare-related nature; for instance, verifying the correct placement of the device before initiating each daily nutrition cycle. Such circumstances call for the involvement of healthcare professionals, rather than mere device management performed exclusively by nursing assistants [24].

This would not simply represent an executive act based on the scheduling of daily activities but would nevertheless require a minimal semiological assessment of the patient. Even in one case out of a hundred, this would imply physical presence and evaluation by the nurse, rather than mere supervision over a group of assisted elderly residents on on-call remote referencing. The weakness, therefore, in the effectiveness of delegation lies precisely in the nursing assistant’s inherent unsuitability to perform healthcare acts autonomously [25]. This, in turn, collides with the extremely complex organizational structure of non-hospital settings, thus entailing risks in terms of patient safety and professional responsibility within the care team, including managerial one [26,27]. It is certainly necessary to take into account the involvement of caregivers and the possibility of adapting feeding methods to what is most feasible in that specific context; however, a certain degree of nursing supervision, expressed through oversight and governance of such activities in long-term care settings, remains essential, thus requiring managers and nurse managers in addressing adequate workforce and expertise staffing. The situation in fact is clearly different compared to the role and support of allied health professionals (AHPs), which can be autonomous [28].

A separate mention should be made of the sensitive issue of therapies, where the roles involved in the entire process, including prescription, preparation, administration, and monitoring, have repeatedly been the reason of medico-legal and judicial evaluations in previous Italian rulings [29]. What emerged, in fact, was a distinction between certain medical functions (e.g., prescribing), nursing functions (e.g., administering treatment), and others that could be delegated depending on the degree of risk and task repetitiveness.

Then, the daily drug administration as far as daily feeding could be delegated, see the oral administration of chronic therapy, while intravenous infusions or intramuscular administrations always depend on prior definite planning and the stability of the patient’s condition, or alternatively on the presence of a nurse for supervision.

However, even in these phases, within residential settings that are less structured than hospital environments, these repetitive and low-discretion activities still rely on the stability of the patient’s condition and on organizational standardization, always under nursing governance and oversight [30]. All these premises, however, lack uniform and consistent implementation in the context under examination; the very one dramatically outlined as the context for the Decree. As a result, non-healthcare personnel become involved in activities that demand particular attention, caution, and verification, at times once again encroaching upon the sphere of clinical and diagnostic assessment (vital signs alerting or therapy adaptation) far beyond their professional competence. This implies a return, or indeed, a rebound, of responsibility onto the healthcare professional. Moreover, it is well known that the context of pharmacotherapy, together with falls, represents one of the processes with the highest frequency of adverse events, considering the high number of medications and supplements prescribed in chronic disease patients, as well as their multiple interactions, by default this constitutes a high risk managerial area [31,32,33].

## 4. Pre-Hospital Emergency Care Context and Protocol-Based Care Models

The topic also specifically concerns the field of pre-hospital emergency care, which, both in the Italian and international contexts, is often staffed by non-healthcare volunteers for first-aid care support. In Anglo-Saxon terminology, these figures are variously and sometimes incorrectly grouped under the term paramedic, despite being differentiated by distinct levels of training and qualification and suffering from both organizational and staffing limitations [34].

However, unlike the Anglo-Saxon context, where such operators are in some cases classified as healthcare professionals, in Italy the possibility for volunteers to carry out acts involving drug administration or interventional procedures remains excluded. Certain therapies, methods of administration, and devices are, in fact, reserved exclusively for medical and intensive care personnel, further restricting the capacity for intervention of non-medical personal to fully medicalized teams. Differently from other countries, AHPs under the Italian regulatory framework cannot prescribe and administer certain therapies autonomously, nor within care plans, as prescription is the exclusive competence of physicians. It should also be emphasized that neither advanced rescuers nor paramedics are authorized to perform any invasive maneuver except under the direct supervision of a physician, both in the German and French systems, in line with the Italian model [35,36].

These activities within pre-hospital emergency settings are conducted in accordance with protocols approved by the medical director of the service, or with the guidelines and protocols developed by the Operations Center for Emergency and Urgent Care and are strictly limited to the procedures explicitly defined therein, within structured care and training programs. To reconcile the needs of multiprofessional integration within healthcare, particularly in performing activities that encroach specific healthcare competencies, the care management plan and the definition of operational protocols were developed. These protocols are structured in clear phases and ordered sequences, with clear assigned “tasks/activities”, roles, and responsibilities, allowing very limited discretionary space and requiring specific and progressive training. This model is widely adopted in pre-hospital emergency settings, relying on emergency scene simulations, critical incident management exercises, effective handovers, feedback systems, and highly standardized operational networks and flowcharts. Such a system partially overcomes the need for the physician’s constant physical presence as in the Anglo-American system, facilitating the performance of collateral healthcare-related activities by non-healthcare personnel, within a predefined framework of delegated tasks, and continuous training and updating.

In this way, roles are clearly distributed and integrated according to a principle of collaboration, mainly through standardized training.

## 5. Profiles of Responsibility

Healthcare activities delegated to NAs could be framed then within the character of pure execution, involving minimal and residual technical or decisional autonomy, and therefore only residual responsibility in implementation.

It should be clarified that, with respect to healthcare professions and their almost universally recognized regulatory requirements, namely a discrete area of competence based on a standardized educational pathway, an autonomous practice through professional board licensing, and adherence to professional ethics, the decisive element in attributing responsibility lies precisely in the absence of autonomy among non-healthcare personnel [37,38].

Consequently, within the scope of nursing care, the delegating professional always retains a residual supervision intended as oversight, while the delegated NA bears a corresponding obligation to promptly and prudently report any deviation from the ordinary course of assigned tasks. This reciprocal responsibility is founded primarily on the clear definition of how delegated activities are to be performed, first and foremost through established care protocols and structured planning, then through effective communication and feedback [39]. The existing research demonstrates a need for interventions to foster a dynamic and effective relationship between nurses and nursing assistants abroad [40]. There is also a need for more interventional studies linking training to improved teamwork, delegation, and communication between the registered nurse and nursing assistant, and and those latter higher expertise to patient outcomes [41].

Under the model set out in the Italian Decree, the nurse, or, analogously, the broad healthcare team in pre-hospital emergency settings, must ensure the application of specific tasks according to the protocols, which specify responsibilities and boundaries of all the providers involved. These protocols and/or procedures, aligned with diagnostic-therapeutic-care pathways and extending to rehabilitation, used for specific care processes (such as stroke, sepsis, oncology, and palliative care), must include, in accordance with evidence, guidelines, and the specific organizational context (primary care, care transition, long-term hospitalization), a concise definition of the required activities, their frequency, assigned roles, and responsibilities, in order to identify those healthcare-related and basic care tasks that can be easily delegated. Therefore, the liability concerns improper, delayed, or omitted execution.

From an organizational perspective, responsibility also—and mainly—arises for the figures charged with implementing and maintaining such systems, both in the administrative sphere, where innovative organizational models must be adopted, and in the technical–healthcare sphere, within the scope of administrative/authoritative organizational responsibility [42]. These professionals are entrusted with developing and maintaining shared protocols, defining responsibilities throughout each phase of the process, and ensuring continuous education and training.

## 6. Conclusions

The multiprofessional approach based on standardization, proceduralizing, and continuous training is undoubtedly more mature in the pre-hospital emergency care setting, where the effective management of complex situations and clear task distribution are well established. However, such experience is still limited in chronic care, palliative care, and especially in long-term and residential facilities, where these new professional figures will be more extensively introduced. In such contexts, technological support for repetitive tasks remains underdeveloped, regional and local policies on staffing vary widely, and there is often a lack of overarching supervisory figures capable of ensuring control of the entire care process or when adaptation in the activities is required. This increases the risk of unauthorized professional practice and reduces overall safety. Consequently, this model can effectively operate within the framework that integrates and innovates staff organization, care and assistance process planning, and continuous training within the context of community-based healthcare.

## Data Availability

The original contributions presented in this study are included in the article. Further inquiries can be directed to the corresponding author.

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
