# Peer review of "Nursing Assistant in Italy: The Principle of Delegation of Health Activities and Liability Profiles"

_nursrep, 2025, doi:10.3390/nursrep15120443_

Round 1

Reviewer 1 Report

Comments and Suggestions for Authors

This paper discusses a relevant issue, the use of less or non-educated assistants in nursing, an urgent topic in a world of a growing global nursing shortage. However, the paper needs to be rewritten more clearly and concisely. Hopefully, my comments will help the authors reach their goal and provide an important message to the nursing profession on this critical issue.

 In summary:

  • Simplify the title
  • Be clear about what this paper is about – purpose of paper and for whom it is written
  • Tell the reader what you are going to present in the paper and why it is of importance to international nursing readers
  • Simply and clarify the text. As put forward it is quite difficult to read and follow, to a point confusing.
  • Re-write the paper.

Title - the title of the paper could be more concise and to the point. A shorter title is more catchy.

Using the term tasks for professional nursing activities is somewhat questionable. Maybe instead of using “task” the authors could refer to “nursing activities” or “nursing elements”. Talking about nursing as task-based indicates that nursing activities are not based on intellectual work and knowledge. I suggest somehow figuring out a way to highlight the knowledge-based work of professional nurses. Delegating assignments to assistants may include carrying out certain tasks, but professional nursing should be knowledge-based.

Abstract – would benefit from revision in terms of language as well as it does not state the purpose / aim of this paper.

  1. Introduction – also here the purpose / aim of paper should be stated at the end of the introduction section.

The text in general - the text is somewhat fragmented. The flow can be improved. I would skip all the multiple paragraph breaks in the narrative. The text is broken up with too many paragraph breaks. Also, the text/language could be more fluid / less complex. For an outsider (not Italian) it is quite difficult to follow the text / narrative and figure out where it is heading.

It would also be helpful to describe the role and responsibility of nurses, and their education, in Italy. What is the distinction between a healthcare professional and a non-healthcare professional in Italy? Licence? Education?

The model set out in the Italian Decree, needs to be explained or displayed early in the paper.

The introduction to what the new role is, lines 117-120, should come much earlier in the paper, for the reader to know what this role includes. Also, it would help the reader to know what training, if any, the NAs have.

It may help to use figures and tables to display the core elements of this new Decree and other issues described in the text.

Comments on the Quality of English Language

Please see previous comments/ suggestions.

Author Response

Reviewer 1 

Authors’ reply 

We have answered the Reviewers’ comments point-by-point reporting on the left side of this table their suggestions and in the right Authors’ answers, with text changes reference. 

In the manuscript, revisions are made with Word track changes. Text additions are written with different colours and underlined, while deletions are shown with strikethrough. 

For improving readability, new sentences or revised paragraph are also highlighted in yellow, as new references in the References section. 

A clear manuscript is provided in pdf format. 

This paper discusses a relevant issue, the use of less or non-educated assistants in nursing, an urgent topic in a world of a growing global nursing shortage. However, the paper needs to be rewritten more clearly and concisely. Hopefully, my comments will help the authors reach their goal and provide an important message to the nursing profession on this critical issue.  In summary: 

  • Simplify the title 
  • Be clear about what this paper is about – purpose of paper and for whom it is written 
  • Tell the reader what you are going to present in the paper and why it is of importance to international nursing readers 
  • Simply and clarify the text. As put forward it is quite difficult to read and follow, to a point confusing. 
  • Re-write the paper. 

Many thanks for your insightful considerations. We have addressed each issue summarized here in the following answers. 

Title - the title of the paper could be more concise and to the point. A shorter title is more catchy. 

We agree. Revised with “Nursing Assistant in Italy: the principle of delegation of health activities and liability profiles” 

Abstract – would benefit from revision in terms of language as well as it does not state the purpose / aim of this paper. 

Revised and integrated with the aim (lines 27-29). 

Using the term tasks for professional nursing activities is somewhat questionable. Maybe instead of using “task” the authors could refer to “nursing activities” or “nursing elements”. Talking about nursing as task-based indicates that nursing activities are not based on intellectual work and knowledge. I suggest somehow figuring out a way to highlight the knowledge-based work of professional nurses. Delegating assignments to assistants may include carrying out certain tasks, but professional nursing should be knowledge-based. 

We are sincerely grateful as we were facing with the difficulty – also linguistic - of distinguishing between delegated activities and individual tasks/duties of the NA. We have revised the text accordingly and organically using activity for the nursing one as deeply relevant. 

Introduction – also here the purpose / aim of paper should be stated at the end of the introduction section. 

The text in general - the text is somewhat fragmented. The flow can be improved. I would skip all the multiple paragraph breaks in the narrative. The text is broken up with too many paragraph breaks. 

We have overall revised introduction, adding aim (lines 127-135), and also answering to other reviewers suggestions on background and comparisons (lines 46-50, 98-115). 

Also, the text/language could be more fluid / less complex. For an outsider (not Italian) it is quite difficult to follow the text / narrative and figure out where it is heading. 

We have revised the text accordingly. 

It would also be helpful to describe the role and responsibility of nurses, and their education, in Italy. What is the distinction between a healthcare professional and a non-healthcare professional in Italy? Licence? Education? 

We have revised the text in the second chapter adding more detail on NAs and nurses in Italy (see lines 187-191). 

The English could be improved to more clearly express the research. 

We have revised accordingly. Sincerely, vb

Reviewer 2 Report

Comments and Suggestions for Authors

The study has merit. The professional nurse and nurse aid must work in unison for best patient outcomes. The act of delegating to the nurse aid is feasible as long as the nurse aid is trained in current procedures and the nurse aids all perform these procedures in the same manner. Since the nurse aid is working under the professional nurse's license, it is imperative that the legalities of this are mandated and that the professional nurse is protected from liability related to the nurse aid performing procedures. 

Author Response

Reviewer 2 

Authors’ reply 

We have answered the Reviewers’ comments point-by-point reporting on the left side of this table their suggestions and in the right Authors’ answers, with text changes reference. 

In the manuscript, revisions are made with Word track changes. Text additions are written with different colours and underlined, while deletions are shown with strikethrough. 

For improving readability, new sentences or revised paragraph are also highlighted in yellow, as new references in the References section. 

A clear manuscript is provided in pdf format. 

The study has merit. The professional nurse and nurse aid must work in unison for best patient outcomes. The act of delegating to the nurse aid is feasible as long as the nurse aid is trained in current procedures and the nurse aids all perform these procedures in the same manner. Since the nurse aid is working under the professional nurse's license, it is imperative that the legalities of this are mandated and that the professional nurse is protected from liability related to the nurse aid performing procedures. 

Many thanks. We have taken the opportunity to improve the content and style of the text following other reviewers suggestions. Sincerely. vb

Reviewer 3 Report

Comments and Suggestions for Authors

The article explores an important topic in the context of healthcare, that is, the need for new roles and interprofessionalisation. This is the case because care needs are augmenting, getting complexier, too, and the current professional roles and human resources are insufficient to respond to current load of needs. This is even truer since the pandemic of COVID 19 and the article highlights how certain sectors of healthcare are faced with even greater scarcity of human resources and professionals, e.g. advanced care for aging adults, residential homes, home care, long term care. The article addresses a specific decree, in Italy, regarding the creation of a new professional, albeit non-clinical role, that of the nursing assistant, to address the “chronic shortage of healthcare personnel”, specifically nurses. This new role is however to support nurses, albeit in low risk contexts and under their close supervision. Yet what the article/authors highlight is that the meaning of this “under supervision” gets muddy when it comes to practice, as well as in specific care contexts, such as long term care or home care. The article explores the tension that exists between theory and practice, whereby the need to have formalized protocols is essential, yet can be faced with various challenges in the concrete contexts of providing care, and ensuring, for example, that the appropriately trained nurse is by the assistant nurse for specific tasks. The article makes the case for an astute attention to given contexts, to ensure that team care is clear, well organized, well structured, adapted to the given context of care. That clear and transparent governance, clear protocols and guidelines, be, also, formulated in ways that are attuned to the given context.

This paper is extremely well written, clear, describes in detail the contexts of care, needing attention, in Italy, with regards to this new professional role. It offers an important view into the relevance but also the implementation care needed for this new professional role, the importance of context adaptation and of clear protocols. I recommend the publication of the manuscript as is; please do revise attentively the English to ensure it is free of typos. I think it may also be of interest for the authors to reflect on how patient safety is always at risk when there is no resource and no one to provide care. While I agree with the authors that clear and robust protocols, guideslines and governance, are needed, I think that too often, when patient safety is discussed and considered, it fails to consider the reverse of the picture, that is : how care services are also responsive when there is no care provided, because there are no resources. In other words: it is not only in actions that responsible ensue, but also in failures to act, in inaction. It may be of interest to readers to see how the authors respond to this important point.

Author Response

Reviewer 3 

Authors’ reply 

We have answered the Reviewers’ comments point-by-point reporting on the left side of this table their suggestions and in the right Authors’ answers, with text changes reference. 

In the manuscript, revisions are made with Word track changes. Text additions are written with different colours and underlined, while deletions are shown with strikethrough. 

For improving readability, new sentences or revised paragraph are also highlighted in yellow, as new references in the References section. 

A clear manuscript is provided in pdf format. 

The article explores an important topic in the context of healthcare, that is, the need for new roles and interprofessionalisation. This is the case because care needs are augmenting, getting complexier, too, and the current professional roles and human resources are insufficient to respond to current load of needs. This is even truer since the pandemic of COVID 19 and the article highlights how certain sectors of healthcare are faced with even greater scarcity of human resources and professionals, e.g. advanced care for aging adults, residential homes, home care, long term care. The article addresses a specific decree, in Italy, regarding the creation of a new professional, albeit non-clinical role, that of the nursing assistant, to address the “chronic shortage of healthcare personnel”, specifically nurses. This new role is however to support nurses, albeit in low risk contexts and under their close supervision. Yet what the article/authors highlight is that the meaning of this “under supervision” gets muddy when it comes to practice, as well as in specific care contexts, such as long term care or home care. The article explores the tension that exists between theory and practice, whereby the need to have formalized protocols is essential, yet can be faced with various challenges in the concrete contexts of providing care, and ensuring, for example, that the appropriately trained nurse is by the assistant nurse for specific tasks. The article makes the case for an astute attention to given contexts, to ensure that team care is clear, well organized, well structured, adapted to the given context of care. That clear and transparent governance, clear protocols and guidelines, be, also, formulated in ways that are attuned to the given context. 

This paper is extremely well written, clear, describes in detail the contexts of care, needing attention, in Italy, with regards to this new professional role. It offers an important view into the relevance but also the implementation care needed for this new professional role, the importance of context adaptation and of clear protocols. I recommend the publication of the manuscript as is; 

please do revise attentively the English to ensure it is free of typos. 

Many thanks. We have taken the opportunity to improve the content and style of the text, also revising typos, following overall reviewers suggestions. 

I think it may also be of interest for the authors to reflect on how patient safety is always at risk when there is no resource and no one to provide care. While I agree with the authors that clear and robust protocols, guidelines and governance, are needed, I think that too often, when patient safety is discussed and considered, it fails to consider the reverse of the picture, that is : how care services are also responsive when there is no care provided, because there are no resources. In other words: it is not only in actions that responsible ensue, but also in failures to act, in inaction. It may be of interest to readers to see how the authors respond to this important point. 

We agree and thank you for the suggestion. We have added a short paragraph in the introduction (lines 46-50). Indeed, there are certainly aspects of omissions-related risk increase and liability; however, under the current framework of Italian law—hence the need for this urgent legislative intervention—what primarily emerges is an issue of unlawful restriction of constitutionally protected right. Sincerely, vb 

Reviewer 4 Report

Comments and Suggestions for Authors

Thank you for the opportunity to review the article "Healthcare providers in the delivery of care: the principle of two delegations of healthcare tasks, care planning, training, and three liability profiles."

This article addresses important and timely issues related to the growing care needs and staffing shortages in nursing, emphasizing the role of the nursing assistant in the therapeutic and treatment process. Despite accurately recognizing the importance of this position in the healthcare system, the study raises some concerns regarding the scope of analysis, clarity, and precision of the information presented.

In the reviewer's opinion, the article focuses almost exclusively on the realities of the Italian healthcare system. Comparative references to the functioning of similar professions—such as medical assistants or community caregivers—in other European countries are lacking, significantly narrowing the perspective and limiting the analytical value of the text. As a result, the study appears fragmented and insufficiently situated in a broader international context. Furthermore, the reviewed text does not sufficiently clearly present the scope of activities delegated to nursing assistants without direct nurse supervision. The lack of such information makes it difficult to assess actual professional independence and the risks associated with performing specific procedures. It also remains unclear what protocols are used to establish the basis for the collaboration between nurses and assistants, as discussed in line 245 – the authors do not indicate their nature or formal basis. The suggestion to change the title of the work also seems justified. The current title suggests a broader issue, while the content is clearly limited to the nurse-assistant relationship. A more precise title would increase transparency and facilitate the reader's proper understanding of the scope of the publication.

In summary, although the article addresses an important and socially significant topic, it requires further refinement in terms of precision, completeness of information, and setting the issue within a broader international context. Only such additions will allow for the fullest utilization of the work's potential and give it greater substantive value.

Author Response

Reviewer 4 

Authors’ reply 

We have answered the Reviewers’ comments point-by-point reporting on the left side of this table their suggestions and in the right Authors’ answers, with text changes reference. 

In the manuscript, revisions are made with Word track changes. Text additions are written with different colours and underlined, while deletions are shown with strikethrough. 

For improving readability, new sentences or revised paragraph are also highlighted in yellow, as new references in the References section. 

A clear manuscript is provided in pdf format. 

Thank you for the opportunity to review the article "Healthcare providers in the delivery of care: the principle of two delegations of healthcare tasks, care planning, training, and the liability profiles." 

This article addresses important and timely issues related to the growing care needs and staffing shortages in nursing, emphasizing the role of the nursing assistant in the therapeutic and treatment process. Despite accurately recognizing the importance of this position in the healthcare system, the study raises some concerns regarding the scope of analysis, clarity, and precision of the information presented. 

In the reviewer's opinion, the article focuses almost exclusively on the realities of the Italian healthcare system. Comparative references to the functioning of similar professions—such as medical assistants or community caregivers—in other European countries are lacking, significantly narrowing the perspective and limiting the analytical value of the text. As a result, the study appears fragmented and insufficiently situated in a broader international context. 

Many thanks. We have taken the opportunity to improve the content and style of the text, following overall reviewers’ suggestions. 

In particular, we have overall revised introduction, adding information on background for some international comparisons (lines 57-61, 98-115). 

Furthermore, the reviewed text does not sufficiently clearly present the scope of activities delegated to nursing assistants without direct nurse supervision. The lack of such information makes it difficult to assess actual professional independence and the risks associated with performing specific procedures. 

We have revised the text in the second chapter adding more detail on NAs and nurses in Italy. 

It also remains unclear what protocols are used to establish the basis for the collaboration between nurses and assistants, as discussed in line 245 – the authors do not indicate their nature or formal basis. 

Many thanks. We have integrated the text with a paragraph mentioning facilities/district care plans with indications of roles and responsibilities of providers, thus highlighting those repetitive, low discretion and executive processes/phases that could be assigned to NA (Lines 373-379) 

The suggestion to change the title of the work also seems justified. The current title suggests a broader issue, while the content is clearly limited to the nurse-assistant relationship. A more precise title would increase transparency and facilitate the reader's proper understanding of the scope of the publication. 

We agree. Revised with “Nursing Assistant in Italy: the principle of delegation of health activities and liability profiles” 

In summary, although the article addresses an important and socially significant topic, it requires further refinement in terms of precision, completeness of information, and setting the issue within a broader international context. Only such additions will allow for the fullest utilization of the work's potential and give it greater substantive value. 

Many thanks. We have taken the opportunity to improve the content and style of the text, following your and other reviewers’ suggestions. Sincerely. vb